# Improving Lipophagy by Restoring Rab7 Cycle: Protective Effects of Quercetin on Ethanol-Induced Liver Steatosis

**DOI:** 10.3390/nu14030658

**Published:** 2022-02-04

**Authors:** Hongkun Lin, Xiaoping Guo, Jingjing Liu, Peiyi Liu, Guibin Mei, Hongxia Li, Dan Li, Huimin Chen, Li Chen, Ying Zhao, Chunjie Jiang, Yaqin Yu, Wen Liu, Ping Yao

**Affiliations:** 1Department of Nutrition and Food Hygiene, School of Public Health, Tongji Medical College, Huazhong University of Science and Technology, 13 Hangkong Road, Wuhan 430030, China; i345575872_99@163.com (H.L.); momodeal@163.com (X.G.); liujingjing690@163.com (J.L.); liupeiyi85@126.com (P.L.); 15695903945@163.com (G.M.); hongxiali201807@163.com (H.L.); lidangoodgirl@163.com (D.L.); hmchen1995@163.com (H.C.); cl707339221@163.com (L.C.); stuzhaoying@163.com (Y.Z.); jcj2010150044@163.com (C.J.); 2Department of inspection and certification, China Certification and Inspection Group Hubei Co., Ltd., Wuhan 430030, China; candis0721@126.com; 3Department of Hepatology, The Second People’s Hospital of Fuyang, Fuyang 236015, China; 4Ministry of Education Lab. of Environment and Health, School of Public Health, Tongji Medical College, Huazhong University of Science and Technology, 13 Hangkong Road, Wuhan 430030, China; 5Hubei Key Laboratory of Food Nutrition and Safety, School of Public Health, Tongji Medical College, Huazhong University of Science and Technology, 13 Hangkong Road, Wuhan 430030, China

**Keywords:** alcoholic fatty liver disease, quercetin, lipophagy, Rab7, TBC1D5

## Abstract

Chronic alcohol consumption retards lipophagy, which contributes to the pathogenesis of liver steatosis. Lipophagy-related Rab7 has been presumed as a crucial regulator in the progression of alcohol liver disease despite elusive mechanisms. More importantly, whether or not hepatoprotective quercetin targets Rab7-associated lipophagy disorder is unknown. Herein, alcoholic fatty liver induced by chronic-plus-single-binge ethanol feeding to male C57BL/6J mice was manifested by hampering autophagosomes formation with lipid droplets and fusion with lysosomes compared with the normal control, which was normalized partially by quercetin. The GST-RILP pulldown assay of Rab7 indicated an improved GTP-Rab7 as the quercetin treatment for ethanol-feeding mice. HepG2 cells transfected with CYP2E1 showed similar lipophagy dysfunction when exposed to ethanol, which was blocked when cells were transfected with siRNA-Rab7 in advance. Ethanol-induced steatosis and autophagic flux disruption were aggravated by the Rab7-specific inhibitor CID1067700 while alleviated by transfecting with the Rab7^Wt^ plasmid, which was visualized by immunofluorescence co-localization analysis and mCherry-GFP-LC3 transfection. Furthermore, TBC1D5, a Rab GTPase-activating protein for the subsequent normal circulation of Rab7, was downregulated after alcohol administration but regained by quercetin. Rab7 circulation retarded by ethanol and corrected by quercetin was further revealed by fluorescence recovery after photobleaching (FRAP). Altogether, quercetin attenuates hepatic steatosis by normalizing ethanol-imposed Rab7 turnover disorders and subsequent lipophagy disturbances, highlighting a novel mechanism and the promising prospect of quercetin-like phytochemicals against the crucial first hit from alcohol.

## 1. Introduction

Alcoholic liver disease is one of the most prevalent liver diseases worldwide, which is caused by harmful alcohol use. Disturbingly, according to the most recent WHO estimates, alcohol abuse was responsible for more than 200 health conditions and over 3 million deaths (5.3% of all deaths) worldwide in 2016 [1,2]. Even with simple and asymptomatic alcoholic fatty liver disease (AFLD), the liver begins to be subjected to burdensome metabolic stress and pathological stimuli, resulting in significant malformations and lesions. Heavy drinkers develop steatosis in 60–100% of cases, steatohepatitis in 20–30%, and fibrosis and cirrhosis in less than 10% of cases [3]. Presently, lipometabolic disturbance induced by alcohol and its metabolites is widely accepted as the “first hit” to initiate hepatic steatosis, and subsequent oxidative stress, inflammatory stimulation, etc., may serve as “second hits”, exacerbating liver susceptibility to advanced pathological conditions [4]. Herein, we aim to explore the potential pathogenesis and effective preventive strategies to block further deterioration of AFLD.

Lipophagy, a subtype of selective macroautophagy for degradation of lipid droplets (LDs) by acid lipase in lysosomes, was first defined as a novel acid lipolysis different from conventional neutral lipolysis in cytoplasm in 2009 [5]. During lysosome-centric lipophagy, cytosolic LDs are engulfed by autophagosomes and transported to lysosomes, where triacylglycerols and other lipids undergo acid lipolysis by lysosomal acid lipase. Though various alcohol models exhibit different autophagy statuses [6,7,8], it is well acknowledged that lipophagy protects the liver from alcohol-induced steatosis [9,10]. However, the molecular mechanism of alcoholic regulation of lipophagy remains largely unclear.

To initiate lipophagy, Rab7, a small GTPase belonging to a large superfamily of Ras-like GTPases, has been presumed to be an essential and crucial organizer, directing membrane fusion between LDs-loaded autophagosomes and lysosomes with the assistance of soluble N-ethylmaleimide-sensitive factor attachment protein receptor proteins (SNAREs) for lysosome degradation and receptor sorting [11]. Previous studies have shown that inhibition of Rab7 activity by alcohol impairs autophagosomes recruitment to LDs [12]. In particular, Rab7 activation was reduced by 80% in the hepatocytes of rats when fed with alcohol for 6 weeks, showing a marked resistance against starvation-induced lipophagy [13,14]. Additionally, prenatal alcohol exposure resulted in the down-regulation of Rab7 in mouse primary brain microvascular endothelial cells [15]. Recently, TBC1D5 (a GTPase-activating protein) has been reported to associate with the Rab7 effector retromer, maintain the normal circulation of the Rab7, and accordingly prevent the accumulation of hyperactivated and immobilized Rab7 around lysosomes [16,17]. However, there is barely any direct evidence indicating that the disorder of the Rab7 cycle through TBC1D5 may mediate alcohol-induced lipophagy dysfunction.

Quercetin (3,3′,4′,5,7-pentahydroxyflavone) is a dietary flavonoid widely distributed in numerous fruits and vegetables and accounts for about 75% of our total flavonol intake [18]. Previous studies have shown that quercetin has anti-inflammatory, anticancer, antidiabetic, hepatoprotective, antibacterial, neuroprotective, and anti-obesity activities [19]. As a direct antioxidant against oxidative stress, quercetin plays a protective role in chronic ethanol-induced liver injury by scavenging ROS, which can reduce the lipid peroxidation products increased by alcohol administration, such as hydroperoxides, MDA, and conjugated dienes [20], thus reducing lipid toxicity. Moreover, a recent systematic review and meta-analysis based on 16 randomized controlled trials has demonstrated that quercetin supplementation can significantly reduce LDL-cholesterol, total cholesterol, and c-reactive protein among patients with metabolic syndrome and related disorders [21]. Our previous research showed that quercetin evidently antagonized ethanol-driven oxidative stress by inducing heme oxygenase-1 via the MAPK/Nrf2 pathway [22], mitochondrial damage, and dyslipidemia [23], and we also preliminarily proved that ethanol can reduce lipophagy in mice [24]. Likewise, epigallocatechin-3-gallate could increase the correlation between Rab7 and LDs in adipocytes [25], and resveratrol restored the distribution of Rab7-positive organelles in cystic fibrosis epithelial cells [26]. However, as far as we know, the regulation of lipophagy by quercetin through Rab7 has not been studied. Thus, the purpose of this study is to investigate the change in membrane fusion mediated by Rab7 during lipophagy in AFLD and to explore whether quercetin plays a protective role in AFLD in maintaining the normal Rab7 cycle.

## 2. Materials and Methods

### 2.1. Antibodies and Reagents

Primary antibodies were purchased from Abcam (antibodies against perilipin 2, PLIN2, Cat. #: ab108323), Santa Cruz (antibodies against GST, Cat. #: sc-138; LAMP2, Cat. #: sc-20011; and TBC1D5, Cat. #: sc-376296), CST (antibodies against Rab7, Cat. #: 9367; LC3II, Cat. #: 2775; and GAPDH, Cat. #: 5174), Protein Tech (antibodies against P62, Cat. #: 18420-1-AP; PLEKHM1, Cat. #: 16202-1-AP; and VAMP8, Cat. #: 15546-1-AP), and Affinity Biosciences (antibodies against SNAP29, Cat. #: DF12743 and Syntaxin17, Cat. #: DF12483). Goat anti-rabbit secondary antibodies (Cat. #: 558002) and goat anti-mouse secondary antibodies (Cat. #: 7076) were obtained from CST. CID1067700 was provided by MCE (HY-13452) and Lipofectamine 3000 was purchased from ThermoFisher Scientific (12566014). DAPI (C1006) and Hoechst (C1002) were bought from Beyotime. IPTG (V3955) and Beads^TM^ GSH (70601-5) were purchased from Promega and Beaver. The plasmids of Rab7^Wt^ (W1949Gn), Rab7^Q67L^ (W1950Gn), Rab7^T22N^ (W1951Gn), CYP2E1, and siRNA-Rab7 were prepared by Gencopoeia.

### 2.2. Animal Model

Male C57BL/6J mice (18–20 g) were purchased from Beijing Vital Rival Laboratory Animal Technology Co., Ltd. The mice model of alcoholic fatty liver disease was established by chronic-plus-single-binge ethanol feeding [27]. Weight-matched mice were randomly divided into four groups: Normal control group (ethanol-free Lieber De Carli liquid diets from Beijing HFK Bioscience Co., Ltd., Beijing, China); Ethanol group (28% ethanol-containing Lieber De Carli liquid diets as an energy source); Ethanol plus quercetin group (28% ethanol-containing Lieber De Carli liquid diets plus quercetin; quercetin, 100 mg/kg·bw); and Quercetin control group. Mice were pair-fed with Lieber De Carli liquid diets containing ethanol or not, and quercetin was dissolved in the liquid diets. After 12 weeks, mice were given oral administration with ethanol (5 g/kg·bw) or maltose solutions for 9 h before being sacrificed. All mice were housed in a temperature-controlled (25 ± 2 °C) environment and kept on a light–dark (12/12) cycle.

### 2.3. Cell Culture and Treatment

Control HepG2 cells were purchased from the American Type Culture Collection (ATCC). HepG2 cells transfected with the CYP2E1 plasmid were cultured in DMEM (Gibco, Grand Island, NE, USA) containing 10% fetal bovine serum, 100 U/mL penicillin (Gibco, Grand Island, NE, USA) and 100 μL/mL streptomycin (Gibco, Grand Island, NE, USA) in a humidified atmosphere with 5% CO_2_ at 37 °C. Every 24 h, the medium was refreshed. Cells were treated for 48 h with different reagents, such as ethanol (100 mM), quercetin (50 μM), or CID067700 (80 μM).

### 2.4. Transfection and RNA Interference

CYP2E1-transfected HepG2 cells were cultured for 12 h (reaching around 70–80% confluence), and then transfected with 100 μM of different siRNA-Rab7 (Table 1) or 1.25 μg/mL of other plasmids (CYP2E1/Rab7^Wt^; CYP2E1/Rab7^Q67L^; CYP2E1/Rab7^T22N^) using Lipofectamine 3000 as indicated by the manufacturer.

### 2.5. Western Blot Analysis

The liver tissues or HepG2 cells were homogenated or lysed with RIPA buffer on ice, and the lysates were collected for Western blot analysis as described in [24]. Bands were detected by the ECL Plus Western Blotting Detection System (GENE GNOMEXRQ, Syngene, UK) and quantified by Image Pro Plus software.

### 2.6. Histological Analysis of Liver Tissues

Liver general pathomorphology was examined by light microscope, and lysosomal ultrastructure, as well as autophagic vacuoles (autophagosomes, autolysosomes, lysosomes containing cytoplasmic material), were further evaluated by the transmission electron microscope (TEM) [24]. In brief, the fresh liver samples (3 mm thick) were fixed with 10% phosphate-buffered formalin (pH 7.4) and embedded in paraffin following dehydration. The paraffin of liver tissue was sectioned into thick sections (5-µm) and stained with hematoxylin and eosin staining (H and E), then observed by the light microscope (Olympus, Japan). For the ultrastructural assay, hepatic samples were fixed in glutaraldehyde (2.5%) to prepare ultrathin sections (80–100 nm) for staining with uranyl acetate and lead citrate, which were imaged by TEM (Tecnai G^2^20 TWIN, Hillsboro, OR, USA).

### 2.7. Double Immunofluorescence and Co-Localization

The paraffin sections of liver tissue were incubated with primary mouse anti-LC3 and rabbit anti-PLIN2 or incubated with primary mouse anti-LC3 and rabbit anti-LAMP2, then incubated with Alexa Fluor 488-conjugated goat anti-rabbit IgG and Alexa Fluor 594-conjugated goat anti-mouse IgG. The images were captured by the fluorescent inverted microscope. HepG2 cells were incubated with LysoTracke red for 30 min before being fixed with 3% formaldehyde and stained with BODIPY_493/503_ (1 μg/mL) for 5 min. Following that, it was washed three times in PBS. After that, the images of lysosomes and LDs were captured by the fluorescent inverted microscope.

### 2.8. Biochemical Analysis

Serum levels of alanine aminotransferase (ALT) and aspartate aminotransferase (AST) were detected by the ALT and AST kits (Nanjing Jiancheng, China). The cholesterol (TC) and triglyceride (TG) levels of livers and HepG2 cells were measured by the TC and TG kits (Applygen, Beijing, China). In brief, liver tissue or HepG2 cells were washed with PBS three times. Then, the lysis buffer was added for liver tissue homogenate or directly to the 12 well plate for cell lysis, standing for 20 min. The lysates were collected into EP tubes. Taking 10 μL of each well for protein quantification by BCA detection kits, the remaining lysates were heated at 70 °C for 10 min, followed by centrifugation at room temperature at 2000× *g* for 5 min, and the supernatant was used for TG and TC determination. Finally, the content of TG and TC was corrected by the protein concentration of each well.

### 2.9. Autophagic Flux Analysis

HepG2 cells were transduced with mCherry-GFP-LC3 lentiviral vectors, which were pre-packed with the lentivirus packaging kit. The quenching properties of the GFP fluorescent signal in acid conditions (pH < 5) and the cherry fluorescent signal do not differ significantly in different acidic environments [28] when used to detect autophagic flux. The image of the LC3 puncta was captured by the fluorescent inverted microscope. After merging the images of green fluorescent and red fluorescent, the intensity of yellow fluorescent (autophagosomes) and red fluorescent (autolysosomes) was determined by image Pro plus software.

### 2.10. Fluorescence Recovery after Photobleaching and Fluorescence Loss in Photobleaching

HepG2 cells that were transfected with the GFP-Rab7 plasmid were cultured in a six-well plate in DMEM (Gibco) containing 10% fetal bovine serum. After 48 h intervention, the cells were passed on glass-bottom culture plates in red-free DMEM (Gibco) supplemented with 10% fetal bovine serum. Then, the fluorescence recovery after photobleaching (FRAP) and fluorescence loss in photobleaching (FLIP) [17] experiments of live cells were performed by the laser-scanning confocal microscope (LSM-I-Live Duo ZEISS LSM 510 DUO and LSM-I-UV ZEISS LSM 510 META), which contains an environmental control system (37 °C, 5% CO_2_), setting pre-bleaching as five images, and using 100% intensity to bleach the selected area by a pulse of the 488-nm laser line and 10 iterations/ROI. The white rectangular regions we selected were used to monitor the photobleaching in FRAP experiments. In FLIP experiments, the yellow rectangular regions we indicated were not bleached by a high-intensity laser but were constantly exposed to a low-intensity laser in the bleaching process to show the loss of fluorescence. The fluorescence images were acquired every second for 100 s during the post-bleaching step. The fluorescence of select regions was monitored and calculated using Leica LAS AF Lite software.

### 2.11. Detection of Active Rab7 by GST-RILP Pulldown Assays

The plasmid of recombinant protein (GST-RILP) was transformed into *E. coli* and expressed. The GST-RILP proteins were collected from bacterial lysate and stored at −80 °C. Then, we used GST agarose beads to band with recombinant protein and used the agarose beads to pull down the active Rab7 that can bind to the Rab7 interacting lysosomal protein (RILP) [29]. Overdose GST-RILP proteins were mixed upside-down with GST agarose beads for 2 h at 4 °C, then used wash buffer for three quick washes. Next, the GST agarose beads connected with GST-RILP were mixed upside-down with cell lysate or liver tissue lysate for 2 h at 4 °C, followed by using elution buffer to treat the agarose beads to retrieve the active Rab7. The active Rab7 protein was separated by 10% SDS-polyacrylamide.

### 2.12. Statistical Analysis

All data were analyzed by SPSS 21 (Armonk, NY, USA). One-way analysis of variance (ANOVA) was used to compare the results, followed by LSD multiple-comparison analysis. Data of animal experiments and cell experiments were expressed as mean ± SD and mean ± SEM, respectively. Set *p* < 0.05 as significance.

## 3. Results

### 3.1. Quercetin Alleviated Liver Injury and Steatosis Induced by Chronic Alcohol Administration

To establish a chronic AFLD model, male C57BL/6 J mice were given a long-term (12 weeks)-plus-one-binge of ethanol. As shown in Figure 1a, chronic alcohol administration led to a disordered arrangement of liver cells and significantly increased LDs deposition and inflammatory infiltration compared with the control group. Following quantitative analysis of the liver injury and hepatic lipid deposition, we found that the levels of serums ALT and AST (Figure 1b,c) and TG content of serum and liver tissue (Figure 1d,e) in the ethanol group were significantly increased compared with the control group. As expected, quercetin intervention effectively normalized the abnormal histopathological changes and biochemical indices induced by ethanol. There was no significant difference between the normal control group and quercetin alone.

### 3.2. Quercetin Restored Hepatic Lipophagy Levels Suppressed by Chronic Alcohol Administration

To investigate how hepatic lipophagy was affected under chronic alcohol administration, paraffin sections of mice liver were subjected to immunostaining for co-localization of LC3 (which exists in the autophagosome and is associated with its development and maturation as an autophagosome monitor) with PLIN2 (the most abundant protein of LDs membrane protein, being assumed as an LDs marker) and LAMP2 (a lysosome membrane protein), to observe the formation of autophagosome and autolysosome. As shown in Figure 1f,g, the co-localization of LC3 with PLIN2 and LAMP2 was reduced in the ethanol group compared with the control group, indicating a lower formation of autophagosomes and autolysosomes. It is well known that the cytoplasmic form of non-lipidated LC3I translates into the membrane-associated lipidated isoform LC3II, plays an important part in autophagosome formation, and P62 serves as a classical autophagosome adapter critical for autophagy function [30]. Therefore, we measured the expression of LC3 and P62 (Figure 1h–k). In comparison with the control group, both the proteins of LC3II and LC3I/LC3II were decreased under alcohol administration. The level of P62 was increased as a consequence of chronic alcohol feeding. The quercetin intervention reversed the results.

### 3.3. Quercetin Restored Membrane Fusion Barrier Caused by Chronic Alcohol Administration

To examine whether chronic alcohol administration caused the disorder of LDs-related autophagosome fusion with lysosomes, a crucial step involving lipophagy, we first performed a TEM to assess the population of autophagosomes, autolysosomes, and lysosomes. As shown in Figure 2a, there were more autophagosomes and enlarged autolysosomes (a defect in autophagosome and lysosome fusion) in alcohol-fed mice, which was normalized by quercetin intervention.

We further examined the expression of main membrane fusion proteins (Appendix A), such as Rab7 (Figure 2b–d), SNAREs (Stx17, SNAP29, and VAMP8 Appendix A) and PLEKHM1 (Figure 2e–g) to explore the changes in autophagosomes and lysosomes fusion in AFLD. The GST-Rab7 interacting lysosomal protein (RILP) pulldown assay showed that GTP-Rab7 was significantly decreased following the ethanol regimen in comparison with normal control, but there was no significant difference in total Rab7. Consistently, the expression of SNAREs was also significantly decreased in alcohol-fed mice. Such down-regulation resulting from ethanol feeding was reversed by quercetin intervention, except for PLEKHM1. Because PLEKHM1 did not alter significantly after ethanol exposure, no additional research into PLEKHM1 in AFLD was conducted.

### 3.4. Rab7 as the Central Molecule of Membrane Fusion Plays a Key Role in Lipophagy

The HepG2 cells were transfected with the CYP2E1 plasmid (Appendix A). Three Rab7 siRNA sequences were used to transfect HepG2 cells (Figure 3a–c), and the results showed that sequences one and three could significantly knockdown Rab7 (knockdown efficiency of 66 and 84%, respectively). Further, we selected sequence three treated cells and co-transfected Rab7 plasmid to investigate the function of Rab7. As shown in Figure 3d–f, knockdown Rab7 cells showed similar lipid deposition and lipophagy blocking to alcohol-treated cells, suggesting that alcohol leads to obstruction of lipophagy through Rab7 and this phenotype was reversed by co-transfected Rab7 plasmid. As shown in Figure 3g–j, there were no significant changes in LC3 protein levels after knockdown of Rab7 compared to control, but P62 levels were significantly increased. This is consistent with the function of Rab7 in the fusion of autophagosomes and lysosomes.

Next, we aimed to explore the relationship between GTP-Rab7 and AFLD. CID1067700 is a Rab7-specific inhibitor, and our findings from the RILP pulldown assay showed that CID1067700 80 μM and above efficiently inhibited the activity of Rab7 (Figure 4a–c). As shown in Figure 4d–f, the expression of GTP-Rab7 significantly decreased after ethanol feeding. CID1067700 and ethanol co-treatment reduced Rab7 activation even more than ethanol alone. The expression of other membrane fusion proteins with the change of Rab7 activation is shown in Appendix A. Following this, we assessed the lipophagy function when inhibiting Rab7 activation, as shown (Figure 4g–j) that P62 further accumulated in the ethanol plus CID1067700 group. The levels of LC3I and LC3II significantly increased in the CID1067700 alone and ethanol plus CID1067700 groups. This is because CID1067700 can efficiently inhibit the degradation of LC3 [31]. To quantify the lipid deposition, we measured the TG and TC levels (Figure 4k,l). In the ethanol and CID1067700 groups, TG and TC levels were increased compared with the control group, and the ethanol plus CID1067700 group further significantly increased the levels of TG and TC. Further, the lipophagy was assessed by the co-location of LDs and lysosomes to show the fusion of lysosomes and autophagosomes, which contain LDs (Figure 4m). The acidophilic dye LysoTracker Red was used to show the lysosomes of living cells, and BODIPY_493/503_ is a bio probe for LDs. The result showed that ethanol treatment caused LDs (green) deposition, and ethanol plus CID1067700 aggravated the situation. Notably, the lysosomes in the ethanol and ethanol plus CID1067700 groups were elongated and showed the dysfunction of lysosomes. Finally, HepG2 cells were transduced with tandem fluorescent-tagged LC3 (mCherry-GFP-LC3) lentiviral vectors to explore the autophagic flux. GFP is acid sensitive and will be quenched in acidic structures (autolysosomes and lysosomes are labeled by only mCherry), while the autophagosomes are indicated by yellow puncta (GFP^+^ mCherry^+^ puncta). The accumulation of yellow puncta shows the autophagy pathway is barred [32]. As shown in Figure 4n–p, in the ethanol and CID1067700 groups, the yellow puncta showed a significant increase, and the fluorescence intensity of the yellow puncta was further increased in the ethanol plus CID1067700 group.

### 3.5. Normal Circulation Rab7 Plays an Important Role in Improving Alcohol-Induced Reduction of Lipophagy

HepG2 cells were transfected with different forms of the Rab7 plasmid (Rab7^Wt^, Rab7^Q67L^, and Rab7^T22N^, respectively) to explore the role of GTP-Rab7 in AFLD. Rab7^Q67L^ as a constitutively active mutation can be mainly GTP-bound. Rab7^T22N^ as a dominant-negative mutation can be mainly GDP-bound. We displayed the immunofluorescence of Rab7 in HepG2 cells that were transfected with a different form of Rab7 (Figure 5a). Consistent with the previous study [33], the fluorescence of Rab7 was associated with perinuclear after overexpression of Rab7^Wt^ and Rab7^Q67L^. However, Rab7 was diffusely distributed in the cytoplasm following overexpression of Rab7^T22N^, which indicated the problem of membrane binding. Further, we assessed the effect of a different form of Rab7 under ethanol treatment. As shown in Figure 5b–e, the expression of LC3II/LC3I was significantly increased and the P62 level was decreased in the Rab7^Wt^ expression ethanol group. The Rab7^Q67L^ expression ethanol group also partly recovered the reduction of autophagy induced by ethanol, but the Rab7^Wt^ expression was more efficient than Rab7^Q67L^, and the Rab7^T22N^ expression did not significantly recover under ethanol treatment. The expression of SNAREs showed similar results (Appendix A). Meanwhile, the TG and TC levels (Figure 5f,g) were consistent with the Western blot results. The overexpression of Rab7^Wt^ and Rab7^Q67L^ before ethanol treatment can efficiently reduce TG and TC levels compared to the ethanol group, but overexpression of Rab7^T22N^ has no significant effect. Next, we assessed the lipophagy levels (Figure 5h) and autophagic flux (Figure 5i–k) after transfecting with different forms of Rab7. The results showed that LDs deposition caused by ethanol treatment was efficiently reduced and the co-location of LDs and lysosomes was increased in the overexpression of Rab7^Wt^ and Rab7^Q67L^ groups. Consistently, overexpression of Rab7^Wt^ and Rab7^Q67L^ reduced the yellow fluorescence induced by ethanol treatment, indicating the recovery of autophagic lipophagy flux. However, overexpression of Rab7^T22N^ appeared to have little protective effect. In addition, the Rab7^Wt^ was more efficient than the Rab7^Q67L^, prompting us to further explore if the normal circulation of Rab7 is more important in restoring ethanol damage.

Therefore, we obtained the protein information interacting with Rab7 through STRING and visualized the protein–protein interaction (PPI) network related to Rab7 through Cytoscape (Figure 2h). The PPI network shows ten nodes focusing on Rab7, and the darker the node is, the closer it is to Rab7. Except for SNAREs (Stx17, VAMP7 and VAMP8), TBC1D5 has the strongest interaction with Rab7. It is confirmed that TBC1D5 is required to inactivate Rab7 (turn GTP-Rab7 to GDP-Rab7), which prevents hyperactivated Rab7 [17]. We measured the expression of TBC1D5 in vivo (Figure 2e,g) and vitro (Figure 6l,m), and the results showed that TBC1D5 was significantly decreased in mouse livers and HepG2 cells treated with ethanol.

### 3.6. Quercetin Targets Rab7 to Play a Protective Role in the Reduction of Lipophagy Caused by Chronic Alcohol Administration

To explore how quercetin reduced the LDs deposition under ethanol treatment, we assessed the lipophagy in the quercetin intervention group (Figure 6a). The results indicated that quercetin efficiently reduced the LDs deposition under ethanol administration and inhibiting the activation of Rab7 resulted in counteracting the protective effect of quercetin. The autophagic flux, as shown in Figure 6b–d, the yellow fluorescence, was decreased by quercetin intervention compared to the ethanol group, but in the CID1067700 and ethanol plus quercetin groups, the flux of autophagy was blocked again. Furthermore, the TG and TC (Figure 6n,o) results showed that quercetin interaction decreased lipid deposition and the decrease was counteracted by CID1067700 treatment. Then, we measured the expression of membrane fusion proteins and autophagy association proteins. As shown in Figure 6e–k, quercetin efficiently recovered the autophagy decreased by ethanol, and the expression of SNAREs (Appendix A) and the activation of Rab7 were significantly increased after quercetin intervention. The CID1067700 treatment almost completely neutralized the protective effect of quercetin. More importantly, quercetin increased TBC1D5 expression in vivo (Figure 2e,g) as well as in vitro (Figure 6l,m). To further investigate the effects of quercetin intervention and alcoholic administration on Rab7 circulation, the fluorescence recovery after photobleaching (FRAP) assays showed that the GFP-Rab7 signal in the control group and quercetin group could rapidly be recovered after photobleaching (Figure 7a,b), whereas the signal in the ethanol group had almost no recovery, indicating that ethanol administration caused a striking loss of Rab7 mobility/membrane turnover. Plus, quercetin intervention could restore the Rab7 mobility/membrane turnover that was disturbed by ethanol administration. Fluorescence loss in photobleaching (FLIP) assays showed the entire cell fluorescence loss during the photobleaching process. This corresponds to the FRAP, as fluorescence loss was minimal in all groups except the ethanol group during photobleaching (Figure 7a,c). In conclusion, quercetin can restore the normal function of Rab7 in both activation and circulation. This provides a new mechanistic basis for the protective role of quercetin in the alcohol-induced lipophagy barrier.

## 4. Discussion

Existing data have indicated a growing prevalence and burden of alcoholic fatty liver disease, particularly end-stage advanced fibrosis, cirrhosis, and carcinoma resulting from excessive alcohol consumption in most of the world’s adult population [34]. Therefore, it is of particular clinical significance for early prevention and effective intervention of AFLD [35]. In the present study, we found that quercetin reversed the damage caused by chronic alcohol administration, which decreased the expression of the membrane fusion proteins (GTP-Rab7, Stx17, SNAP29, and VAMP8) and normalized the lipophagy level. More importantly, we found the expression of TBC1D5 was downregulated after alcohol administration and quercetin intervention restored the expression of TBC1D5 and the Rab7 cycle. We demonstrated that quercetin exerted a protective effect on AFLD by restoring the normal circulation of Rab7 to improve lipophagy.

AFLD is associated with lipid metabolism disorders and insulin resistance, which lead to alcoholic steatosis [36]. Alcohol consumption decreases the ratio of NAD+/NADH, which leads to fatty acid synthesis increase (upregulates SREBP1c in the liver [37]) and decrease in fatty acid transport and oxidation (downregulates PPARα [38] and mitochondrial β-oxidation [39]) as the “first hit”, and chronic alcoholism inhibits lipophagy, though may activate with compensatory acute alcohol administration against alcoholic liver injury [40,41]. Lipid accumulation occurs when lipophagy, a conserved protective manner of the body, is impaired [42]. The mechanism of lipophagy dysfunction in AFLD is still not fully elucidated, but several molecules and pathways involved have been studied. For example, TFEB and Dyn2, responsible for lysosomes recovery and biogenesis, were inactivated in alcohol-fed rats and alcohol-induced hepatitis patients [43,44,45]. In addition, chronic alcohol administration increases the intrahepatic leucine, which further enhances the activity of mTORC1, showing autophagy inhibition [46,47]. Meanwhile, the fusion of autophagosomes and lysosomes is impaired by chronic alcohol administration, and Rab7, as the key regulator of autophagosome-lysosome fusion, plays an important role in this process [33]. A previous study showed that primary hepatocytes from rats fed an alcohol diet for 6 weeks were insensitive to starvation-induced lipophagy [13]. Consistently, we showed that chronic alcohol administration decreased autophagic flux, which leads to lipophagy dysfunction in the liver, due to a decrease in Rab7 activation. However, Eid et al. showed that chronic alcohol administration increased the autophagosomes in rat livers. The difference may be due to the difference in detection methods and measurement criteria of autophagic flux.

Quercetin exists mostly in the form of quercetin glycosides, which are found in fruits and vegetables [48]. Quercetin glycosides are hydrolyzed by β-glucosidases and extensively metabolized in enterocytes and the liver [49]. Quercetin inhibits low-density lipoprotein oxidation by suppressing myeloperoxidase-catalyzed oxidation and plays an anti-inflammatory, antioxidation, and lipid-lowering role in AFLD by decreasing serum and hepatic NF-κB, p65, COX-2, IL-6, and TG levels [23,50,51,52]. It is reported that quercetin mediates upregulation of HO-1 through the ERK/Nrf2 pathway to reduce the production of ROS, thereby exerting a protective effect against AFLD [23,53]. A recent study in a zebrafish model showed that quercetin can downregulate the P2X7R through the PI3K/Keap1/Nrf2 pathway to protect liver function in AFLD [54]. In addition, our previous research showed that quercetin decreased the activated AMPK and PLIN2 levels and increased the co-location of LC3II and PLIN2 to promote lipophagy in AFLD [24]. In the present study, we proved that quercetin plays a protective role in AFLD by restoring the normal circulation of Rab7 to restore lipophagy and also indicated that quercetin increased the expression of TBC1D5, which controls the inactivation of Rab7 under alcohol administration. However, the mechanisms of how quercetin restores the Rab7 cycle need to be further explored.

Rab7, as a regulator of vesicle transport, is important for many cellular processes. For instance, in lipophagy, mitophagy, apoptosis, protein sorting, maturation of the endosome, fusion of the lysosome and autophagosome, and lysosome biogenesis [11,16,55]. The immunoblot analysis indicated that Rab7-like Ypt7p associated with LDs, Vma13p, and V1 parts of the vacuolar (H^+^) ATPase (V-ATPase) as the partners of Ypt7p coordinated the LDs’ dynamic and membrane trafficking [56]. This indicates that Rab7 plays a central regulatory role in lipophagy. The function of Rab7 is guaranteed by a cyclical mechanism to switch between activation (GTP-bound state) and inactivation (GDP-bound state). Consistent with previous studies [13,14], we found that chronic alcohol administration did not change the expression of total Rab7 but could significantly reduce the activation of Rab7. Other studies have shown that alcohol treatment can reduce the expression of total Rab7 [15,57]. This may be due to differences in models, duration, and dose of intervention. However, we indicated that overexpression of Rab7^Q67L^ is less effective than Rab7^Wt^ in reducing lipid deposition. Similarly, previous studies indicated that Rab7^Q67L^ showed lower GTPase activity compared to Rab7^Wt^, and Rab7^Q67L^ also dramatically reduced the turnover of lysosomes [58,59]. All of this suggests that normal Rab7 cycling is more important for liver health than Rab7 activation.

The most important regulator of cyclical Rab7 is Guanine Nucleotide Exchange Factors which are required for GTP capture and GDP dissociation, and GTPase-Activating Proteins which are required for GTP hydrolysis. When lacking retromer-bound TBC1D5 (one of the GTPase-Activating Proteins), Rab7 cannot sharply localize and accumulate over the lysosomes in an active state. Therefore, active Rab7 cannot be released from the lysosome so that it is unavailable locally activated elsewhere [17]. To explore why Rab7^Wt^ can better recover the level of lipophagy that was decreased by alcohol administration, we measured the expression of TBC1D5, and the results showed chronic alcohol administration significantly reduces the expression of TBC1D5, which indicates alcohol not only reduces Rab7 activation but also causes the cyclical mechanism to be disturbed. To our knowledge, there is no previous study on the regulation of quercetin on Rab7 and TBC1D5, so our findings further illustrate how quercetin improves lipophagy and exerts AFLD protection by restoring the Rab7 cycle. In addition, since the circulation of Rab7 plays an important role in maintaining the normal function of the membrane contact sites (MCSs) [29], the formation defect of the MCSs caused by Rab7 will be further explored in the next steps with AFLD.

## 5. Conclusions

We indicated that quercetin recovers the Rab7 mobility/membrane turnover, which is disturbed by ethanol administration to normalize lipophagy, thus reducing the lipid deposition in AFLD. Notably, the normal circulation of Rab7 plays a more important role in maintaining liver function than its activation alone. Quercetin is now known as a phytochemical remedy for the treatment of a variety of hepatic diseases, and the current study highlights it as a nutritional supplement with promising prophylactic and therapeutic prospects.

## Figures and Tables

**Figure 1 nutrients-14-00658-f001:**
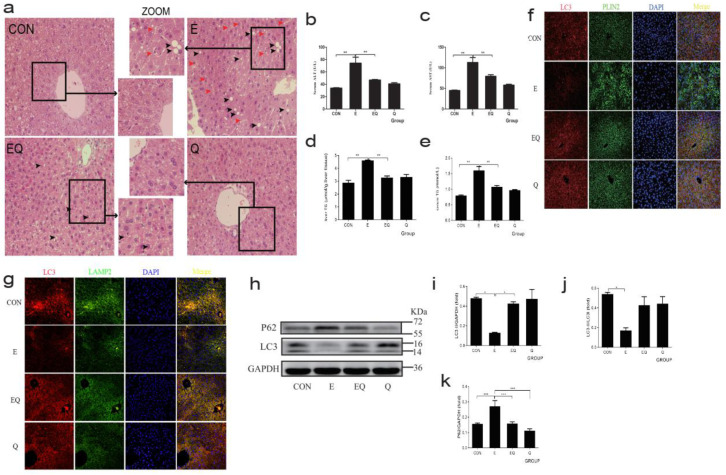
Quercetin alleviated steatosis and restored lipophagy in the liver induced by chronic alcohol. (**a**) Hematoxylin and eosin-staining of mice fixed liver tissue section, observed by light microscope (400×). The black arrow indicates LDs, the red arrow indicates inflammatory infiltration. (**b**–**e**) Serum ALT and AST, liver TG and serum TG. (**f**) The immunostaining of paraffin sections of mice liver for co-localization of LC3 and PLIN2. (**g**) LAMP2 and LC3 were observed by fluorescent microscope (400×). Representative images were shown. (**h**) Western blot analysis was performed to measure proteins of LC3 and P62 (*n* = 6). (**i**–**k**) GAPDH was used as a protein loading control, and the results were quantified in three independent experiments per condition, with densitometry analysis of (**h**). Values were shown as means ± SD (*n* = 6). * *p* < 0.05, ** *p* < 0.01, *** *p* < 0.001. CON: normal control group; E: ethanol group; EQ: ethanol plus quercetin group; Q: quercetin group.

**Figure 2 nutrients-14-00658-f002:**
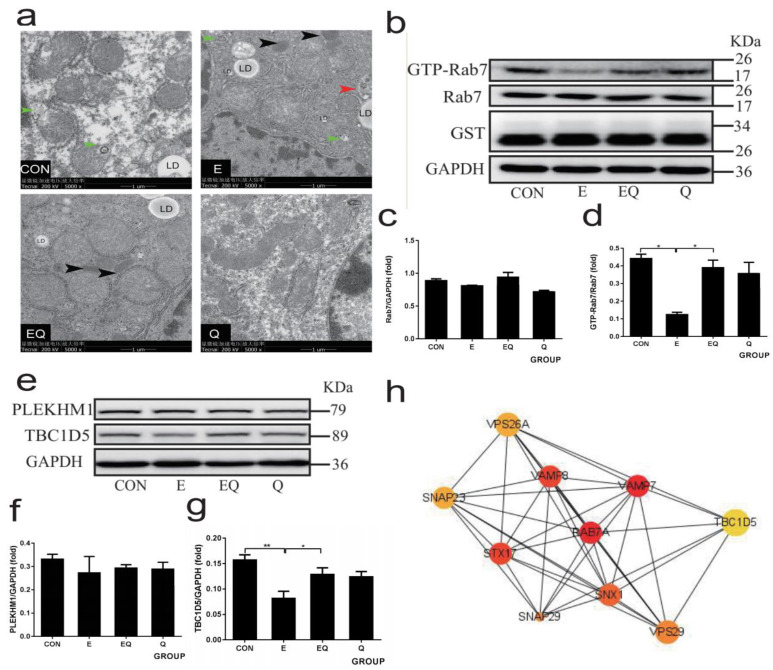
Effects of quercetin on lipophagy and membrane fusion in mice treated with chronic alcoholism. (**a**) Transmission electron microscope (TEM) images showed autophagic vesicles in cells from different groups. Black arrowheads show lysosomes, red arrowheads show autolysosome, green arrowheads show autophagosomes. LD show lipid droplets. (**b**) Western blot of GTP-Rab7 (obtained by GST-RILP pulldown) and total Rab7. (**c**,**d**) Densitometry analysis of (**b**). (**e**) Western blot of Rab7-associated proteins PLEKHM1 and TBC1D5. (**f**,**g**) Densitometry analysis of (**e**). GAPDH was used as a protein loading control, and the results were quantified in three independent experiments per condition. (**h**) A PPI network of membrane fusion proteins centered on Rab7, ten nodes were shown, the deeper color and larger size indicating a stronger correlation with Rab7. Data shown are means ± SD (*n* = 6). * *p* < 0.05, ** *p* < 0.01.

**Figure 3 nutrients-14-00658-f003:**
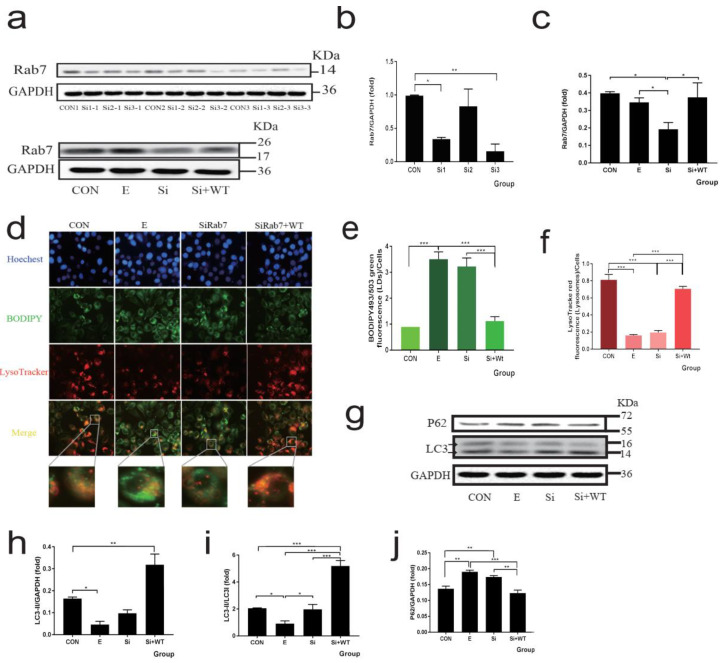
Rab7 is required for LDs de-creation through lipophagy in alcohol administration in HepG2 cells. (**a**) Western blot of Rab7 after transfection with three siRNA-Rab7 or Rab7 plasmid. (**b**,**c**) Densitometry analysis of (**a**). (**d**) Co-localization of lysosomes (LysoTracke-red) and LDs (BODIPY_493/503_) observed by inverted fluorescence microscope (400×). (**e**,**f**) Fluorescence intensity of LDs (green) and lysosome (red). (**g**) Western blot of LC3 and P62, GAPDH was used as a protein loading control, and the results were quantified in three independent experiments per condition. (**h**–**j**) Densitometry analysis of (**g**). Data shown are means ± SEM (*n* = 3). * *p* < 0.05, ** *p* < 0.01, *** *p* < 0.001. CON: normal control group; E: ethanol group; si: small interfering RNA group; si + WT: small interfering RNA plus wild type Rab7 plasmid group.

**Figure 4 nutrients-14-00658-f004:**
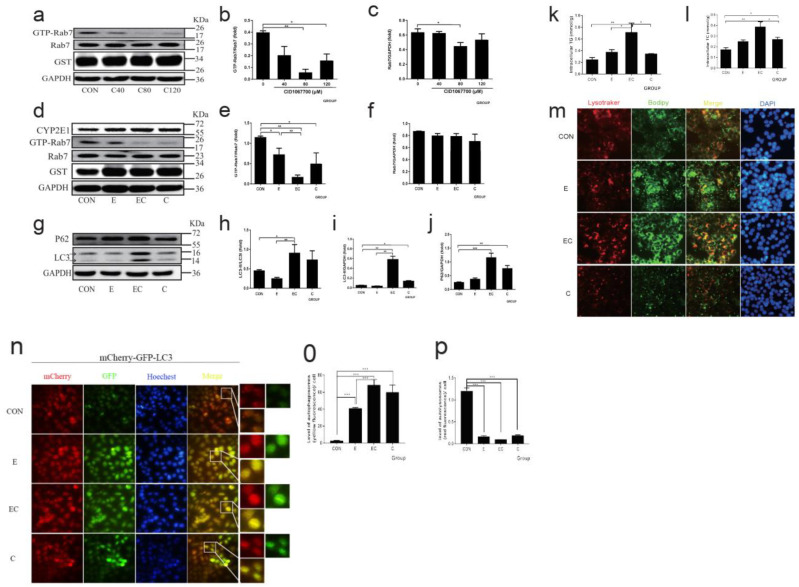
Pharmacological inhibition of Rab7 aggravates alcoholic-induced decreased lipophagy in HepG2 cells. (**a**) Western blot of GTP-Rab7 (obtained by GST-RILP pulldown) and total Rab7 of HepG2 cells treated with 40 μM, 80 μM, and 120 μM CID1067700, respectively. (**b**,**c**) Densitometry analysis of (**a**). (**d**) Western blot of GTP-Rab7 (obtained by GST-RILP pulldown) and total Rab7. (**e**,**f**) Densitometry analysis of (**d**). (**g**) Western blot of LC3 and P62. (**h**–**j**) Densitometry analysis of (**g**). The GAPDH was used as a protein loading control, and the results were quantified in three independent experiments per condition. (**k**) TG of HepG2 cells. (**l**) TC of Hepg2 cells. (**m**) Co-localization of lysosomes (LysoTracke-red) and LDs (BODIPY_493/503_) observed by inverted fluorescence microscope (400×). (**n**–**p**) Live-cell imaging of HepG2 cells expressing mCherry-GFP-LC3 to explore the autophagic flux observed by the inverted fluorescence microscope (400×). Data shown are means ± SEM (*n* = 3). * *p* < 0.05, ** *p* < 0.01, *** *p* < 0.001. CON: normal control group; E: ethanol group; EC: ethanol plus CID1067700 group; C: CID1067700 group.

**Figure 5 nutrients-14-00658-f005:**
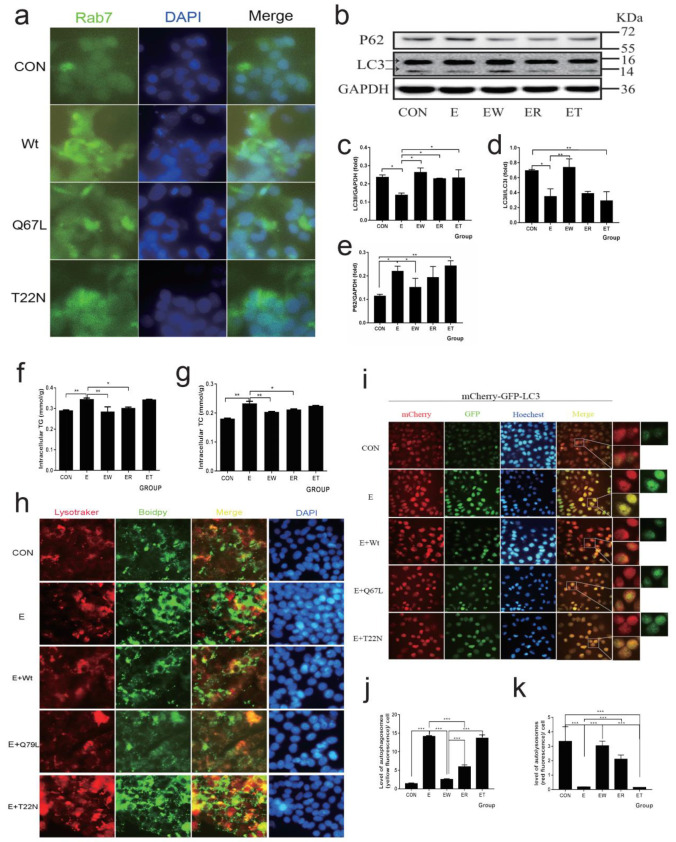
The effects of different forms of Rab7 on lipophagy. (**a**) Immunostaining of Rab7 for Hepg2 cells that were transfected with different active forms of the Rab7 plasmids (WT, Q67L, T22N) was observed by the inverted fluorescence microscope (400×). (**b**) Western blot of LC3 and P62. GAPDH was used as a protein loading control, and the results were quantified in three independent experiments per condition. (**c**–**e**) Densitometry analysis of (**b**). (**f**) TG of Hepg2 cells. (**g**) TC of Hepg2 cells. (**h**) In Hepg2 cells that were transfected with different active forms of the Rab7 plasmid (WT, Q67L, T22N), co-localization of lysosomes (LysoTracke-red) and LDs (BODIPY_493/503_) was observed using the inverted fluorescence microscope (400×). (**i**–**k**) Live-cell imaging of Hepg2 cells expressing the mCherry-GFP-LC3 to explore the autophagic flux observed by the inverted fluorescence microscope (400×). Data shown are means ± SEM (*n* = 3). * *p* < 0.05, ** *p* < 0.01, *** *p* < 0.001. CON: normal control group; E: ethanol group; EW: Hepg2 cells expressing Rab-Wt plus ethanol; ER: Hepg2 cells expressing Rab-Q67L (active form) plus ethanol; ET: Hepg2 cells expressing Rab-T22N (inactive form) plus ethanol.

**Figure 6 nutrients-14-00658-f006:**
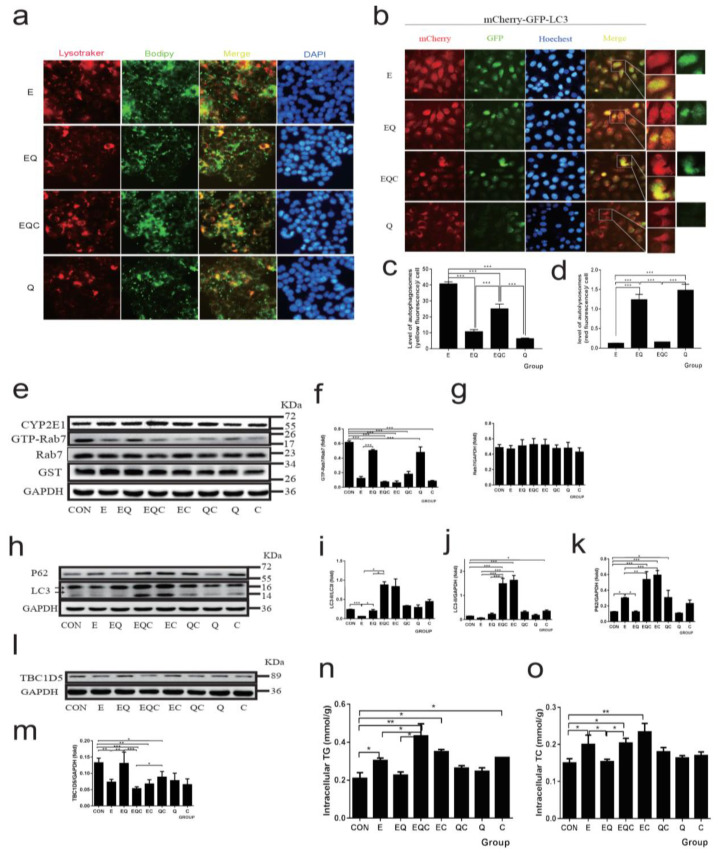
Effect of quercetin on lipophagy and autophagy flux by regulating Rab7 in Hepg2 cells. (**a**) Co-localization of lysosomes (LysoTracke-red) and LDs (BODIPY_493/503_) was observed by an inverted fluorescence microscope (400×). (**b**–**d**) Live-cell imaging of Hepg2 cells expressing the mCherry-GFP-LC3 to explore the autophagic flux observed by the inverted fluorescence microscope (400×). (**e**) Western blot of GTP-Rab7 (obtained by GST-RILP pulldown) and total Rab7. (**f**,**g**) Densitometry analysis of (**e**). (**h**) Western blot of LC3 and P62. (**i**–**k**) Densitometry analysis of (**h**). (**l**) Western blot of TBC1D5. (**m**) Densitometry analysis of (**l**). The GAPDH was used as a protein loading control, and the results were quantified in three independent experiments per condition. (**n**) TG of Hepg2 cells. (**o**) TC of Hepg2 cells. Data shown are means ± SEM (*n* = 3). * *p* < 0.05, ** *p* < 0.01, *** *p* < 0.001. CON: normal control group; E: ethanol group; EQ: ethanol plus quercetin group; EQC: CID1067700 an ethanol plus quercetin group; EC: ethanol plus CID1067700 group; QC: quercetin plus CID1067700 group; C: CID1067700 group Q: quercetin group.

**Figure 7 nutrients-14-00658-f007:**
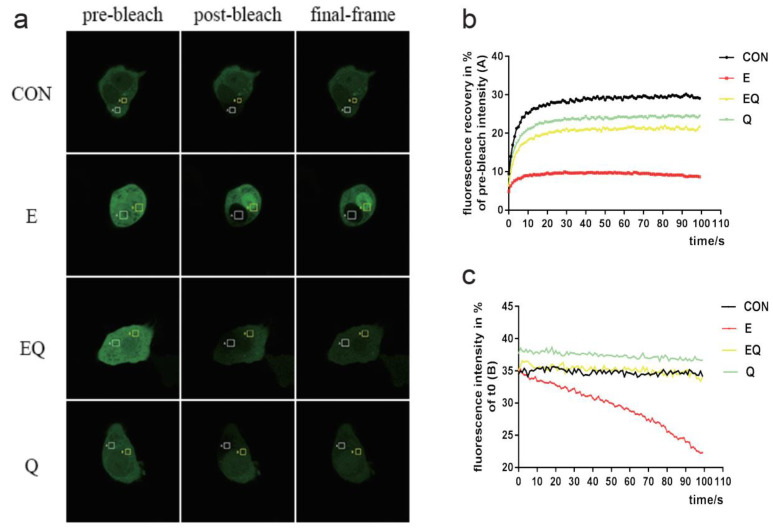
Effect of quercetin on Rab7 mobility/membrane turnover. (**a**,**b**) The GFP-Rab7 plasmid was transfected into HepG2 cells and used FRAP imaging in live cells to monitor its mobility/membrane turnover (in area A, as indicated) (*n* = 4). (**a**,**c**) The GFP-Rab7 plasmid was transfected into HepG2 cells and used FLIP imaging in live cells to monitor its mobility/membrane turnover (in area B, as indicated) (*n* = 4). CON: normal control group; E: ethanol group; EQ: ethanol plus quercetin group; Q: quercetin group.

**Table 1 nutrients-14-00658-t001:** Sequences of 3 siRNA-Rab7.

	Primer Name	Sense (5′-3′)	Antisense (5′-3′)
si1	Human-RAB7-1	CCACAAUAGGAGCUGACUUTT	AAGUCAGCUCCUAUUGUGGTT
si2	Human-RAB7-2	GCUAGUCACAAUGCAGAUATT	UAUCUGCAUUGUGACUAGCTT
si3	Human-RAB7-3	CCAGACGAUUGCACGGAAUTT	AUUCCGUGCAAUCGUCUGGTT

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
