# Peer review of "Improving Lipophagy by Restoring Rab7 Cycle: Protective Effects of Quercetin on Ethanol-Induced Liver Steatosis"

_nutrients, 2022, doi:10.3390/nu14030658_

Round 1

Reviewer 1 Report

In this study, the authors suggested that the quercetin restores Rab7 circulation which is decreased by ethanol, thus quercetin normalizes the lipophagy and induces degradation of lipid droplets.  

Experimental methods are well organized. However, there are some critical points that should be addressed. Based on data presented, some interpretations must be amended or justified.

  1. The authors explain that ethanol reduces the level of LC3.

1-1. However, in Figure 1f, the intensity of total LC3 in the ethanol group was not decreased compared to control, whereas other LC3 results in figure 1g-i showed a decrease. Please provide the quantification result of Fig. 1f and sufficient explanation about this result.

1-2. In figure 3, LC3-II in Fig 3g appeared not to be reduced in the ethanol-treated group compared with control, however, its quantification result in Fig 3h showed that LC3-II was decreased by 0.5-fold than the control. The authors are asked to explain this apparent discrepancy.

1-3. In the Fig. 4n, tandem fluorescent-tagged LC3 were increased in EtOH treated group compared to the control. Please provide more detail explanation about result of LC3 expression level. 

  1. In line 239-241, author explain “P62 servers as a classical autophagosome adapter critical for autophagy function.” Please provide the reference. Also, in the previous study, LC3II and P62 increase in direct proportion in P62-mediated lipophagy [1]. The results of this study showed that LC3II and P62 were expressed overall in inverse proportion. Please provide sufficient explanation regarding to this discrepancy.

  1. The authors mentioned that CYP2E1 transfected HepG2 cells were used. Were Transfected HepG2 cells used for all in vitro studies? If not, please provide this information in each figure legend or result. Also, why do the authors use CYP2E1 transfected HepG2 cells? Please provide the explanation.

  1. Based on authors description, the attenuation of RAB7 activity by CID1067700 lead to lipid accumulation. However, in Fig 4m, the bodipy which indicate lipid droplet in the only CID 1067700 treated cells appeared to be decreased than control. Please provide the explanation and the quantification result of Fig 4m.

  1. Signs of significance between E and EQ are missing in Fig 1j and k.

  1. The explanation about PLEKHM1 were missing. Please provide the sufficient information about PLEKHM1

  1. Is GTP-Rab7 is the activated form of Rab7? Please provide the information.

  1. In Figure 3d, the label is represented as “E+siRAN”. Please clarify what is correct; “E+siRAB” or only “siRAB”. Also, please clarify “siRAB+WT” or “E+siRab7+WT”.

  1. Please provide the merged image with Hoechst or DAPI in Fig3-6.

  1. Is LD means lipid droplet? Please provide a full name of this abbreviation where mentioned firt.
  2. Is ALD means alcoholic liver disease? Please provide a full name of this abbreviation, and also confirm and unify ALD or AFLD.

  1. In Figure 4a, the abbreviations of C40, C80 and C120 should be explained in the figure legends.

  1. In Fig 5d, please confirm the y-axis. That axis should be labeled “LC3II/LC3I”, not “LC3I/LC3II”.

Reference mentioned in this comment.

1          Lam T, Harmancey R, Vasquez H, Gilbert B, Patel N, Hariharan V et al. Reversal of intramyocellular lipid accumulation by lipophagy and a p62-mediated pathway. Cell Death Discov 2016; 2: 16061.

Reviewer 2 Report

The authors provided the first evidence showing that quercetin was effective in ameliorating ALD by restoring Rab7 cycle. The major issues were low ALD severity in the ethanol group (fail to appropriately establish the ALD animal model) and no error bars in multiple figures (e.g., Figure 1i, 1j 1k, 3e, h, j), meaning that the results of the control group were almost exactly the same, which was not possible. So the authors need to provide the original blots of all replicates. Also, the authors need to re-analyze the data by incorporating Tukey HSD since multiple comparisons were performed.

Minor points:

Line 58: do you mean alcohol-mediated endotoxin production or transportation?

The authors should provide the full name of the acronyms when they appeared the first time (such as LDs)

Method: what is the rationale of using male mice?

What is the number of replicates (biological and technical) of Histology analysis?

Line 131: Is it one-time ethanol gavage? Why did administered alcohol before animal sacrifice?

What’s the passage number of the HepG2 cells that you used for analysis?

Please provide brief information on how you extracted TC and TG from the HepG2 cells.

Line 203: E. Coli

The resolution of Figure 1A is poor. Inflammatory infiltrations were not clearly annotated. The authors need to provide the area of the liver covered by lipid droplets. ALD severity seems to be comparable across the groups.

 Figure 1g: the intensity of DAPI in the Quercetin group was much lower than in other groups.

Round 2

Reviewer 1 Report

No further comments

Reviewer 2 Report

I appreciate the authors' effort in addressing my questions and the quality of the manuscript was significantly improved. However, the authors still need to revise their method in analyzing their western blots. You don't need to adjust for the protein expression in the control group, and just presenting the ratio of your targeted protein and GAPDH in EACH group should do the work. Since there were variations within the control group, the lack of SEM bars in the control group is confusing and erroneous.  

The role of DAPI in ICC acts as an internal standard, just as GAPDH to western blot, so a unanimous intensity of DAPI across the interventional groups is necessary. Please revise your ICC figures. 

Also, please include the TG and TC extraction methods in your Method Section of the main text.

Round 3

Reviewer 2 Report

The authors have successfully addressed my questions and made sound modifications.